# The Antifungal Antibiotic Filipin as a Diagnostic Tool of Cholesterol Alterations in Lysosomal Storage Diseases and Neurodegenerative Disorders

**DOI:** 10.3390/antibiotics12010122

**Published:** 2023-01-09

**Authors:** Francesco Bruno, Serena Camuso, Elisabetta Capuozzo, Sonia Canterini

**Affiliations:** 1Regional Neurogenetic Centre (CRN), Department of Primary Care, ASP Catanzaro, 88046 Lamezia Terme, Italy; 2Association for Neurogenetic Research (ARN), 88046 Lamezia Terme, Italy; 3Division of Neuroscience, Department of Psychology, Sapienza University of Rome, 00185 Rome, Italy; 4Department of Biochemical Sciences, Sapienza University of Rome, 00185 Rome, Italy

**Keywords:** filipin staining, polyene antibiotics, diagnosis, cholesterol, neurodegenerative diseases, lysosomal storage diseases, Alzheimer’s disease, Niemann-Pick type C disease, Huntington disease, GM1 gangliosidosis

## Abstract

Cholesterol is the most considerable member of a family of polycyclic compounds understood as sterols, and represents an amphipathic molecule, such as phospholipids, with the polar hydroxyl group located in position 3 and the rest of the molecule is completely hydrophobic. In cells, it is usually present as free, unesterified cholesterol, or as esterified cholesterol, in which the hydroxyl group binds to a carboxylic acid and thus generates an apolar molecule. Filipin is a naturally fluorescent antibiotic that exerts a primary antifungal effect with low antibacterial activity, interfering with the sterol stabilization of the phospholipid layers and favoring membrane leakage. This polyene macrolide antibiotic does not bind to esterified sterols, but only to non-esterified cholesterol, and it is commonly used as a marker to label and quantify free cholesterol in cells and tissues. Several lines of evidence have indicated that filipin staining could be a good diagnostic tool for the cholesterol alterations present in neurodegenerative (e.g., Alzheimer’s Disease and Huntington Disease) and lysosomal storage diseases (e.g., Niemann Pick type C Disease and GM1 gangliosidosis). Here, we have discussed the uses and applications of this fluorescent molecule in lipid storage diseases and neurodegenerative disorders, exploring not only the diagnostic strength of filipin staining, but also its limitations, which over the years have led to the development of new diagnostic tools to combine with filipin approach.

## 1. Cholesterol Metabolism in the Brain

Cholesterol (C27H45OH) is the most abundant member of a family of polycyclic compounds known as sterols, and represents an amphipathic molecule, such as phospholipids, with the polar hydroxyl group located in position 3 and the remaining part of the molecule is completely hydrophobic. In cells, it is usually present as free, unesterified cholesterol, or as esterified cholesterol, in which the hydroxyl group binds to a carboxylic acid and thus generates an apolar molecule [1] (Figure 1).

The brain, even if it represents only 2% of the body mass, represents the organ that contains more cholesterol than the rest of our body [2], in which it plays a key structural and functional role since is it involved in the development of neurons, neurite growth, synapse formation, transmission, and plasticity [3]. In addition, cholesterol represents an essential constituent of plasma membrane [4] and the cytoplasmic precursors of several key molecules, such as vitamin D, oxysterols, bile acids, and steroid hormones [5,6,7]. Given these countless pleiotropic functions, an imbalance of cholesterol could potentially compromise proper brain function. De novo cholesterol synthesis, therefore, emerges in neurons, astrocytes, and oligodendrocytes over early postnatal neurogenesis, after evolving most significantly in astrocytes that supply it to neurons. In particular, the 3-hydroxy-3-methylglutaryl-CoA reductase (HMGR) enzyme produces cholesterol in astrocytes, and it is controlled by a negative feedback mechanism via the sterol-regulated element binding protein (SREBP), which binds to sterol-regulated element 1 (SRE-1) in the promoter of the HMGR gene [8,9,10]. The cholesterol is then combined with apolipoprotein E (APOE), the biggest apolipoprotein found in the brain, and released by ABC transporter family member A1 (ABCA1), also known as cholesterol efflux regulatory protein (CERP) [11,12]. The low-density lipoprotein receptor (LDLR) is then bound by the APOE-cholesterol complex, which is subsequently taken up by neurons [11,12]. The complex is then transported within late endosomal/lysosomal (LE/Ly) compartments, where two proteins, NPC1 and NPC2, will mediate the efflux of cholesterol from these organelles [13].

The mechanisms by which The NPC1 and NPC2 proteins assist in the release of cholesterol from endosomes and lysosomes are still being studied. The sequential action of these two proteins has been attributed to two potential mechanisms [14]. The NPC2 protein, which is located in the lumen LE/Ly compartments, binds to the cholesterol contained in the vesicles, according to the first model (Model A). NPC1 then gets cholesterol from NPC2 and sends it across the membranes dividing LE and Ly, producing the cholesterol available to other cellular compartments, such as the ER, PM, and mitochondria [2,15]. In particular, the transfer of cholesterol from NPC2 protein to NPC1 protein requires the reorientation of the N-terminal domain (NTD) of NPC1, a key protein domain for cholesterol binding [16]. Mutations in NPC1-NTD inhibit the cholesterol translocation from NPC2 to NPC1, leading to reduced lysosomal export of cholesterol [17,18,19]. The second model (Model B) states that NPC1 localizes on the membrane surface of LE/Ly and releases cholesterol from LE/Ly vesicles to NPC2. Both theories enlist the assistance of an additional, as-yet-unidentified protein that shuttles cholesterol away from the LE/Ly membrane and toward other locations [12].

## 2. The Antifungal Antibiotic Filipin

Filipin is an antifungal polyene macrolide antibiotic characterized by a large lactone ring containing five conjugated double bonds [20] isolated from the bacterium *Streptomyces filipinensis* [21,22,23]. It is a naturally fluorescent compound that binds to cholesterol, but not to esterified sterols, and, therefore, is commonly used as a marker for labelling and quantifying free cholesterol in cells and tissues [24,25,26]. In fact, while in the past it has been used as a probe in the study of the membrane structure and sterols distribution [27], more recently it is employed to evaluate the cholesterol content in membranes and organelles [28]. Chemically, filipin is a blend of four components: Filipin I (4%), II (25%), III (53%), and IV (18%), which should more correctly be referred to as the Filipin complex [29].The hydrophobic interaction between filipin and cholesterol is achieved thanks to the presence of the 3β-OH group in the chemical structure of cholesterol [30], which generates a “filipin-cholesterol” complex located in the hydrophobic layer of biological membranes [31]. Interestingly, this complex induces a rupture of cell membranes, which progressively leads to the release of cytoplasmic components [32], like glucose and enzymes, as observed by electron microscopy in rat erythrocytes [33]. In fact, since filipin perturbs the structure of lipidic bilayer when it binds to cholesterol, this staining can only be performed in fixed cells and tissues to avoid the appearance of aggregates in the plasma membrane where the filipin-cholesterol complex is located [34] (Figure 2).

Although filipin has been used for years as an indicator of both membrane and intracellular cholesterol levels, this labelling has important limitations, such as the sensitivity of filipin to photobleaching, which may limit its use, especially in intracellular cholesterol quantification experiments [26,28]. To overcome these limitations, Wilhelm et al. [28] developed a protocol that involved the use of a next-generation confocal microscope equipped with a 355 nm UV laser and ultra-sensitive detectors to prevent photobleaching of the filipin. Furthermore, by treating live cells with methyl-β-cyclodextrin, a compound capable of reducing the cholesterol level from biological membranes [35], the authors were able to visualize and quantify only the intracellular cholesterol content by filipin staining [28].

Despite these disadvantages, filipin staining still represents a promising diagnostic tool for understanding of dyshomeostasis of cholesterol present in several brain diseases [36,37,38]. Here, we comprehensively summarize and review the use of filipin as a histochemical marker for unesterified cholesterol both in neurodegenerative (e.g., Alzheimer’s Disease and Huntington’s disease) and lipid storage diseases (e.g., Niemann-Pick Type C Disease and GM1 gangliosidosis).

## 3. Filipin as a Diagnostic Tool for Lysosomal Storage Diseases

Lysosomal storage diseases (LSDs) are a group of inherited metabolic disorders in which harmful amounts of lipids, glycoproteins, or mucopolysaccharides accumulate in various cells and tissues because of a deficiency of a single enzyme required for their metabolism [39]. Among the LSDs, the main ones are the Niemann-Pick type C disease and GM1 gangliosidosis [40,41]. A recent line of evidence suggests considering unesterified cholesterol as a central player in both of these two pathologies [42,43], prompting researchers to better characterize the use of filipin staining as their biomarker.

### 3.1. Niemann Pick Type C Disease

Niemann-Pick type C (NPC) disease represents an autosomal recessive lysosomal storage disorder [44,45] caused by mutation of the *NPC1* or *NPC2* genes [2,46,47] and characterized by a wide spectrum of clinical manifestations. Since NPC1 and NPC2 proteins intercede the exit of cholesterol from endosomes/lysosomes, mutations in the genes encoding for these proteins results in an impairment of unesterified cholesterol egress in LE/Ly compartments [42,48,49,50]. Consequently, cholesterol and other lipids (e.g., sphingolipids, glycolipids, and lysosome-specific phospholipids) tend to accumulate in these compartments [14,51,52,53,54,55,56,57].

Thus, the defects in lipids trafficking in NPC disease cause premature death due to hepatosplenomegaly and progressive neurodegeneration [46]. Albeit the NPC1 and NPC2 complementation groups are indistinguishable biochemically and clinically, it is commonly called NPC1 and NPC2 disease in the literature [58]. However, the high heterogeneity in the clinical manifestations of NPC disease hinders the diagnosis [59]. For these reasons, the diagnosis of NPC disease can take a very long time, even years, and must be assisted by specialists in various disciplines [60,61,62].

Wraith and collaborators of the NPC Guidelines Working Group suggested performing a clinical investigation and medical reconstruction, as well as laboratory tests, on individuals suspected of having this disease. In particular, the authors proposed a specific laboratory algorithm that includes bone marrow aspiration or biopsy to investigate the presence of foam cells (*not mandatory*) and the analysis of the serum chitotriosidase, which generally showed elevated activity in NPC patients (*not mandatory*). Finally, authors proposed the skin biopsy (*mandatory*) to isolate fibroblasts on which it is necessary to examine the presence of cholesterol alterations [61], preferentially with the filipin test [58]. Using filipin staining and a fluorescence microscope, it was shown that in in vitro cultured fibroblasts from NPC1 patients, the unesterified cholesterol concentration was higher than controls, suggesting an altered cellular distribution of cholesterol resulting in its accumulation within LE/Ly [63,64]. According to the recommendations of the NPC Guidelines Working Group [61], from the filipin test four different results can be obtained: (i) *clearly negative*—which a priori excludes the presence of NPC; (ii) *highly positive*—this “classic” picture is shown by 80–85% of NPC patients and reveals the nearly sure presence of the disease; (iii) moderately positive with pure LDL—this “variant” picture is shown by approximately 15% of NPC patients and reveals the presence of a probable variant of the disease; and, (iv) *difficult to interpret*—this concerns 3–5% of NPC patients and requests the re-examination of the clinical characteristics and, if deemed necessary, the sequencing of the NPC1 gene to detect the presence of possible mutations [65].

However, conclusive diagnosis of NPC needs the presence of mutations in the NPC1 or NPC2 genes [45,58,61,65,66,67]. Thus, Patterson et al. [65,68] suggested carrying out in parallel filipin staining and genetic analysis with the aim to obtain complementary information, even if they recognized both as diagnostic tests of choice for NPC disease. Moreover, genetic analysis is also recommended for prenatal diagnosis for couples with previously affected children, even in case of filipin-negative test results [69]. On the other hand, filipin staining is also a valid instrument to estimate the functional value of new NPC1 or NPC2 genetic variants, as well as to confirm diagnosis in heterozygous patients [58].

Interestingly, Takamura et al. [70] also demonstrated that the minimally invasive filipin test in blood smears of NPC patients could be considered as a good screening valuable tool for distinguishing NPC disease from other LSDs (i.e., Fabry disease, Gaucher diseases, Neuronal Ceroid Lipofisicinosis, and Niemann-Pick disease type A and B), though no direct correlations were detectable between fluorescence intensity and the clinical stages of NPC patients. Conversely, Hammerschmidt et al. [71] reported that plasma analysis of cholestane-3β,5α,6β-triol, the major cholesterol metabolite, has a higher specificity and sensitivity for the diagnosis of NPC disease, compared to the filipin staining of fibroblasts in NPC patients, opening new ways for the less invasive diagnosis of this disease. In addition, Vanier and Latour [72] reported a difficult interpretation of filipin staining in 15% of an NPC patients’ cohort. Sometimes several NPC1 mutations, such as p.P1007A, are associated with mild alterations in cholesterol trafficking and thus with a variant filipin profile [58]. These more recent data question the usefulness of continuing to use this elective historical method for diagnosing NPC disease.

### 3.2. GM1 Gangliosidosis

GM1 gangliosidosis (GM1) is a heterogenous lysosomal storage disorder, due to mutations in the Galactosidase Beta 1 gene (*GLB1*), with an age of onset ranging from infantile period to adulthood [73]. This mutation leads to a defective activity of the enzyme β-galactosidase (βgal) that causes the accumulation of GM1 ganglioside within brain, resulting progressively in neurodegeneration, represented by generalized paralysis, severe emaciation, and death [74].

Preliminary data indicated the presence of filipin fluorescence in fibroblasts of GM1 gangliosidosis patients [25,43] and lysosomal membrane of brain tissue of βgal^−/−^ mice [75], suggesting an accumulation of unesterified cholesterol along with GM1 in this disease. However, Arthur et al. [76] demonstrated that the accumulation of GM1 was not associated with a significant increase of cholesterol content in microsomal membranes of brain tissues of βgal^−/−^ mice. These results highlight the role of the filipin as a marker of GM1, as well as cholesterol, in GM1 mouse models. In addition, this finding could explain why the treatment with cyclodextrin abrogates filipin signal in fibroblast derived by patients affected by NPCD and mouse brain tissues [17,77,78], but has no effect in GM1 patients and mice [77]. On the contrary, it opens a controversy on the use of filipin staining as a selective marker of unesterified cholesterol.

## 4. Filipin as a Diagnostic Tool for Common Neurodegenerative Diseases

Neurodegenerative diseases, such as Alzheimer’s Disease and Huntington’s disease, include a diversified group of conditions characterized by a progressive cells degeneration of the central and/or peripheral nervous system [79]. A recent line of evidence suggests considering cholesterol metabolism dyshomeostasis as a candidate mechanism underlying the pathogenesis of these two pathologies [80,81], leading researchers to characterize better the use of filipin staining as their biomarker.

### 4.1. Alzheimer’s Disease

Alzheimer’s disease (AD) is the most common form of dementia, primarily characterized by cognitive, functional, and neuropsychiatric symptoms [82,83]. AD can be classified into familial (FAD), rare and at early onset, and sporadic, the major form with a late onset [84]. The FAD form is due to mutations in one or more of the following genes: amyloid precursor protein (*APP)*, presenilin-1 (*PS1*), and presenilin-2 (*PS2*) [84,85].

The major clinical hallmarks of AD are extracellular deposits of β-amyloid peptide (Aβ), resulting from the proteolytic cleavage of the amyloid precursor protein (APP), and intracellular accumulation of tau protein, due to its hyperphosphorylation [86,87]. Interestingly, studies conducted on brain samples of postmortem subjects with AD have demonstrated the involvement of cholesterol metabolism in AD pathogenesis [88]. In detail, growing evidence supports the idea that cholesterol metabolism and cholesterol oxidation products contribute to beta-amyloid plaque formation, tau hyperphosphorylation, and that cholesterol oxidation products concur to AD neurodegeneration and pathogenesis [80,89], although this process is not entirely clear.

Several pieces of evidence have shown an increase in cholesterol levels in the frontal gyrus of AD patients and defects in brain cholesterol metabolism [90,91]. By filipin staining on brain sections of AD patients, it was understood that the senile plaques are enriched with both cholesterol and apolipoprotein E (ApoE), the cholesterol transporter that to date is considered as the main gene candidate for AD onset [92,93].

Likewise, studies conducted by Montesinos and collaborators on a mouse model of AD, have demonstrated an increase in 99-aa C-terminal fragment of APP (C99) level, derived from a higher cleavage of APP by β-secretase [94]. They demonstrated a toxic accumulation of C99 in mitochondria-associated ER membranes (MAM), a key region for lipid homeostasis regulation. They observed, by filipin staining, that the C99 fragment had a cholesterol-binding domain and the abnormal concentration of C99 at MAM induced an increase in cholesterol esterification by acyl-coenzymeA: cholesterol acyltransferase 1 (ACAT1), in an AD mouse model [94]. Moreover, the pathological accumulation of C99 within MAM triggers an exacerbated uptake of extracellular cholesterol and increases its trafficking from PM to MAM, leading to lipid dyshomeostasis and neurodegeneration [94].

Furthermore, the contribution of cholesterol to AD pathogenesis seems to be correlated with the interaction between the Aβ peptide and cholesterol. Consistently, in vivo, and in vitro studies exhibited that cholesterol dyshomeostasis triggers the activity of β and γ secretase, two enzymes involved in APP processing, exacerbating the production of Aβ peptides [95,96].

Membrane cholesterol has been identified as the main link between aging, Aβ plaques, and tau pathology in neurons. Indeed, Nicholson et al. [97] demonstrated that in the presence of hippocampal Aβ, functionally mature neurons were more susceptible to tau cleavage and cell death than young and immature neurons. Furthermore, this neuronal change in susceptibility to Aβ toxicity emerged concurrently with an increase in cholesterol levels on neuronal membranes.

As mentioned above, the accumulation of cholesterol oxidation products contributes to AD onset. Chief among these oxidation products are two oxysterols known as 24S-hydroxycholesterol and 27-hydroxycholesterol [98], which can cross the BBB, in contrast to cholesterol. To demonstrate this, it was observed that the post-mortem brains of patients exhibited a decrease in 24S-hydroxycholesterol content, but also a significant increase in 27-hydroxycholesterol levels [99,100]. Furthermore, in AD patients, plasma levels of 24S-hydroxycholesterol appear to be increased during the early phases of the disease [101,102] and the blood quantification of filipin levels in peripheral mononuclear cells revealed an increase in fluorescence related to cognitive decline [80].

Since Aβ synthesis and aggregation also cause lipid dysregulations, the analysis of their levels could be able to predict the risk of conversion from Mild Cognitive Impairment (MCI) to AD. In particular, it has been proposed that the alteration of cholesterol metabolism begins in the presymptomatic stages of the disease, progressively worsening on the onset of cognitive deficits. Based on this evidence, the filipin test could therefore be useful to differentiate MCI from AD and to understand the real risk of patients with MCI to progressing to AD [80,103].

### 4.2. Huntington Disease

Huntington’s disease (HD), a neurodegenerative disease with an autosomal dominant pattern of inheritance, results from an expansion of CAG repeats in the IT15 gene encoding the Huntingtin (HTT) protein required for axonal transport [104,105]. Neuropathologically HD is characterized by severe cortical atrophy, striatal degeneration, and astrogliosis [106,107], resulting in motor impairment, cognitive decline, psychiatric symptoms, and death by about 15–20 years from its onset [108]. A common feature in an increasing number of studies in HD animal models, cell lines, and various patient samples is the overall downregulation of cholesterol metabolism (for a review see: [81,109,110], prompting researchers to consider cholesterol as a key player also in this pathology.

It is well known that in HD the mutant huntingtin protein (mHTT) inhibits the caveolin 1-mediated endocytosis of cholesterol in striatal neurons [111]. Caveolin-1 is a protein capable of binding cholesterol and helping it in its intracellular trafficking [112]. Using filipin staining, it was demonstrated that striatal neurons expressing mHTT exhibit an abnormal accumulation of cholesterol related to an impaired trafficking of cholesterol mediated by caveoline-1 [111]. However, conflicting results were achieved in these studies. Some research groups reported an increase [111,113,114] in cholesterol levels, whereas others a decrease [115,116,117,118], or no changes in cholesterol from both animal models and human patients [109,119]. As evidenced by Marullo et al. [120], these mixed findings could be attributable to the different experimental techniques and materials employed in the several quantitative analyses for cholesterol levels. In the attempt to resolve this controversy, the authors made a comparison among several methods—enzymatic methods, colorimetric methods (filipin staining and Nile Red staining), and gas chromatography-mass spectrometry (GC-MS)—to measure the cholesterol levels in HD cell models. The results of this study showed that, in HD samples, the identification of cholesterol by colorimetric and enzymatic assays should be associated with the use of much more sensitive analytical methods.

## 5. Conclusions

Despite its relative invasiveness, filipin staining still seems to be a useful method for identifying cholesterol anomalies in NPC and AD patients, even if it seems increasingly appropriate to combine this colorimetric investigation with genetic testing, especially in the case of negativity to filipin despite the documented presence of mutations in other family members. Promising data also indicate that the less invasive blood quantification of filipin levels represents a potential predictors of AD risk, whereas the minimally invasive filipin test in blood smears of NPC patients could be considered as a good screening tool useful for distinguishing NPC disease from other LSDs. Other research is needed to corroborate this evidence. Conversely, the evidence that, in GM1 gangliosidosis, filipin also marks GM1, and that GC-MS appears to be a more appropriate method for identifying cholesterol alterations in HD, raises doubts about its specificity and opens the way towards the study and characterization of new more precise cholesterol biomarkers, both in neurodegenerative and lysosomal storage diseases. Moreover, recent preliminary evidence [121] indicated the presence of cholesterol alterations in other neurodegenerative diseases, such as Frontotemporal dementia [122], and lipid storage disease, such as α-mannosidosis [123]. Future research should investigate the diagnostic potential of filipin also in the fibroblasts and brain tissues of these patients.

## Figures and Tables

**Figure 1 antibiotics-12-00122-f001:**
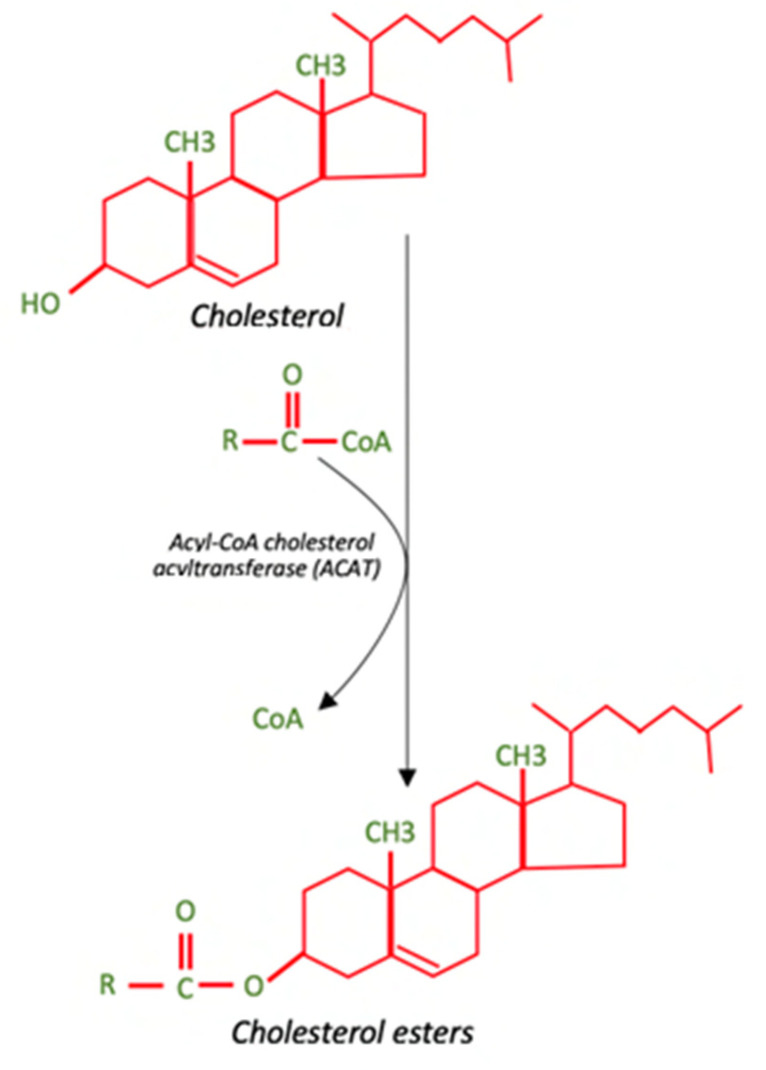
Chemical structure of cholesterol and cholesterol esters.

**Figure 2 antibiotics-12-00122-f002:**
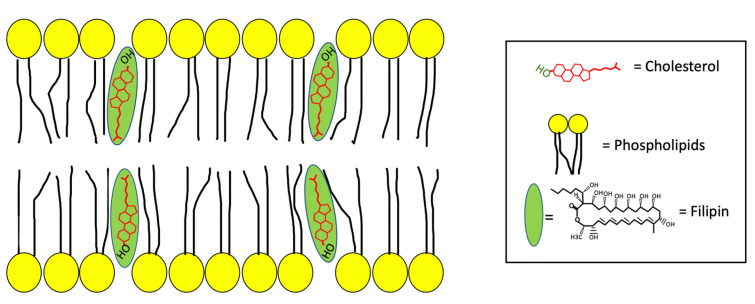
The binding of filipin to cholesterol in the plasma membrane.

## Data Availability

Not applicable.

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
