# Peer review of "The Antifungal Antibiotic Filipin as a Diagnostic Tool of Cholesterol Alterations in Lysosomal Storage Diseases and Neurodegenerative Disorders"

_antibiotics, 2023, doi:10.3390/antibiotics12010122_

Round 1

Reviewer 1 Report

In this review the authors gather all available data on the use of filipin, a fluorescent molecule known to bind free cholesterol, as a marker for lysosomal storage diseases and in the neurodegenerative diseases Alzheimer and Huntington. They display the, sometimes contradictory results from various studies and emphasize on different points: the possibility of non-invasive filipin staining in blood smear for the differential diagnostic of NPC disease, the non-specific staining of GM1 by filipin in GM1 gangliosidosis and the systemic lipid (including cholesterol) dysregulation prior to symptom onset in Alzheimer Disease. The review is very well structured. It brings together results from a lot of different sources and the data described is discussed thoroughly, combining a critical view on the usefulness of filipin staining in some cases with suggestions on how to extend the use of this molecule in other cases. Finally, the topic addressed is of interest to both, fundamental researchers working on lipids/cholesterol as well as people involved with the medical applications of cholesterol detection.  

Author Response

Thank you to the reviewer for this comment. We have now corrected the typos (marked in red).

Reviewer 2 Report

Review article by Francesco Bruno et al, discussed the pro and cons of diagnostic methods used in lysosomal storage disorders and neurodegenerative disorders based on cholesterol alterations of the antifungal antibiotic filipin in this review article. 

The manuscript writing is good however there are certain changes needs to be made. I would like to accept the manuscript with minor changes as the review article is useful content and interesting .

Below are suggestions for minor revision of the manuscript:

1.     Article section 3.1 line 150 “any how the impairment” of lipid trafficking needs to be replaced with appropriate words.  

2.     Line 183 the word “in any case nowadays” would be better to replace with alternate word.  

3.     Conclusion section line 326 sentence “more over to until dementia sentence can be moved to other parts of the manuscript instead of conclusive section.

I have no major concerns with this paper to publish in antibiotics Journal after minor revision

Author Response

appropriate words. 

Response. Thanks to the reviewer for this suggestion. We have now replaced “any how the impairment” with “Thus, the defects in lipids trafficking” (marked in red).

Comment 2. Line 183 the word “in any case nowadays” would be better to replace with alternate word. 

Response. Thanks to the reviewer for this suggestion. We have now reformulated the sentence in this way: “However, the definite diagnosis of NPC requires the presence of mutations in NPC1 or NPC2 genes” (marked in green).

Comment 3. Conclusion section line 326 sentence “more over to until dementia sentence can be moved to other parts of the manuscript instead of conclusive section.

Response.  This sentence is in the conclusions because it opens up future prospects: to examine in further studies whether filipin can be considered a good biomarker also for other neurodegenerative and lysosomal storage diseases. In accordance with the editor's comment, we have expanded this sentence by also adding the a-mannosidosis (marked in red).